# THREE FORMS OF STOCHASTIC INJECTION FOR IMPROVED DISTRIBUTION-TO-DISTRIBUTION GENERATIVE MODELING

## ABSTRACT

Modeling transformations between arbitrary data distributions is a fundamental scientific challenge, arising in applications like drug discovery and evolutionary simulation. While flow matching offers a natural framework for this task, its use has thus far primarily focused on the noise-to-data setting, while its application in the general distribution-to-distribution setting is underexplored. We find that in the latter case, where the source is also a data distribution to be learned from limited samples, standard flow matching fails due to sparse supervision. To address this, we propose a simple and computationally efficient method that injects stochasticity into the training process by perturbing source samples and flow interpolants. On five diverse imaging tasks spanning biology, radiology, and astronomy, our method significantly improves generation quality, outperforming existing baselines by an average of 9 FID points. Our approach also reduces the transport cost between input and generated samples to better highlight the true effect of the transformation, making flow matching a more practical tool for simulating the diverse distribution transformations that arise in science.

## 1 INTRODUCTION

Modeling transformations between arbitrary distributions is a canonical problem in science. Consider drug discovery, where the challenge is to find compounds capable of transforming diseased cells into a healthy state. This task is complicated by the inherent heterogeneity of cell populations, meaning the desired states must be treated as a distribution. Furthermore, observational constraints like destructive assays yield unpaired 'before' and 'after' snapshots, making a one-to-one mapping impossible (Zhang et al., 2025; He et al., 2024). This fundamental challenge — learning a distributional transformation from unpaired data — extends across scientific fields, from understanding disease progression in patients to tracing the evolution of galaxies over cosmological timescales (He et al., 2025; Anstine & Isayev, 2023; Wu et al., 2025; Höllmer et al., 2025).

Flow matching offers an elegant framework for distribution-to-distribution learning. Unlike diffusion models, which typically transport Gaussian noise into the data distribution, flow matching can directly model a transformation between two arbitrary empirical distributions (Lipman et al., 2022; Liu et al., 2022). Despite this theoretical promise, it has primarily been leveraged for learning noise-to-data. We study the more general, scientifically relevant, yet underexplored data-to-data setting, and diagnose a critical failure mode: sparse supervision. With finite samples from *both* source and target distributions, training supervision is available only along one-dimensional interpolant trajectories that sparsely cover the sample space. Consequently, the learned velocity field that overfits these few interpolations, leading to poor generalization. Our controlled experiments on a synthetic problem (Section 2.2) confirm that the performance of flow matching drastically deteriorates with increasing data dimensionality or decreasing number of training data.

To counteract sparsity, we propose a simple and effective intervention: inject three forms of stochasticity during training to densify the supervision signal. First, we propose a two-stage training scheme inspired by transfer learning, initially training on the less data-sparse task of mapping noise to target before fine-tuning on the source-to-target transformation. Second, we perturb the source samples with Gaussian noise, augmenting the available data to a denser source distribution. Third, we perturb

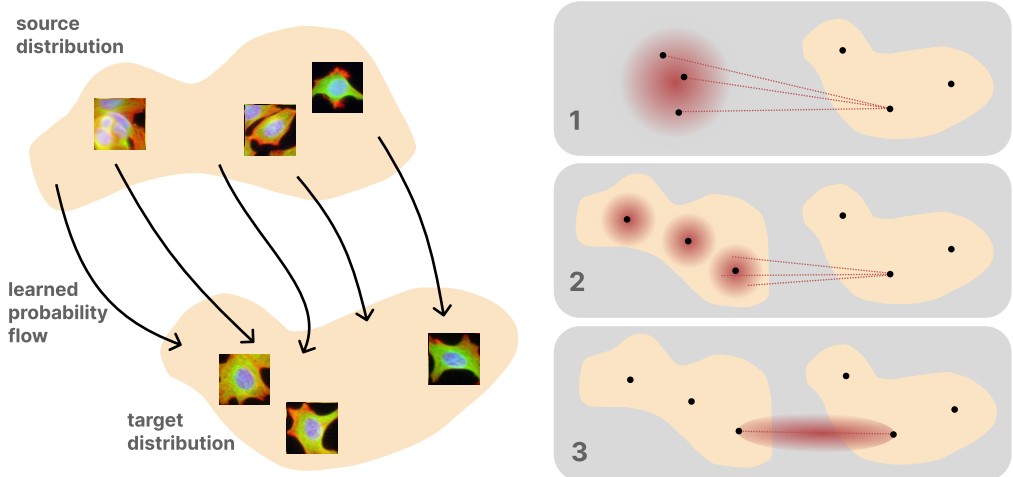

Figure 1: Our objective is to learn a flow from source onto target distributions, given unpaired training samples. Left: In the illustrated example, we learn to simulate cellular response to a chemical intervention. Right: We introduce stochastic injections that alleviate the sparsity challenges of distribution-to-distribution learning from finite target *and* source training examples, by 1) transfer learning from the noise-to-target task, 2) perturbing source samples, and 3) perturbing the training interpolant.

the flow interpolant with a noise schedule, creating a denser set of interpolating points between each source-target pair sampled during training. Our theoretical analysis supports these perturbations' ability to alleviate sparsity and improve generalization. Figure 2 provides complementary intuition: the stochastic injections induce more space-filling interpolants than standard flow matching.

We validate our method on five challenging, high-dimensional image datasets covering natural and scientific domains, spanning problems in biology, radiology, and astronomy. The tasks range from modeling cellular response to chemical treatments (Caie et al., 2010), to simulating the effects of cosmological redshift (Do et al., 2024). Our stochastic injections significantly improve generation quality, outperforming vanilla flow matching by 13 FID (Frechet Inception Distance) points and other established baselines by 9 FID points. Moreover, our stochastic injections improve the transport cost between a given source sample and its generated counterpart in the target distribution, as measured by Euclidean distance in pixel space. This means there is a closer visual correspondence between source and generated target, which better highlights the true effect of the distributional transformation at a sample level.

In summary, we identify flow matching as a promising solution to the scientifically important problem of distribution-to-distribution learning, but find that the standard formulation struggles in high dimensions due to data sparsity. We propose stochastic injections that alleviate sparsity and improve generalization, while being simple to implement, computationally cheap, and compatible with ODE sampling. The resulting recipe turns flow matching into a practical tool for learning unpaired distribution-to-distribution transformations, advancing data-driven modeling for scientific discovery.

## 2 PROBABILITY FLOWS AND DATA SPARSITY

In this section, we review probability flows and show, using controlled experiments on a synthetic task, how *data sparsity* impedes standard flow matching in the distribution-to-distribution setting.

### 2.1 PRELIMINARIES

Consider data distributions living on a metric space $\mathcal{X}$, such as $\mathcal{X} = \mathbb{R}^d$. The objective of generative modeling with flow matching is to learn a time-dependent velocity field $v_t(x)$ such that each sample $x_0 \sim p_0$ from the source distribution is smoothly transported by $v_t(x)$ onto a sample $x_1 \sim p_1$ from

the target distribution by the ODE

$$\mathrm{d}x_t = v_t(x_t)\mathrm{d}t, \quad x_0 \in p_0, x_1 \in p_1, t \in [0, 1]. \tag{1}$$

The most common setting, which we call *one-sided* distribution matching, is interested in learning only the data distribution $p_1$.[1] In this case it is standard to choose $p_0 = \mathcal{N}(0, 1)$ (Lipman et al., 2022; Liu et al., 2022). In this work, we consider the more general and scientifically relevant *two-sided* setting, where both $p_0$ and $p_1$ are non-trivial distributions from which we have access to a finite number of training examples. In the case of drug discovery, $p_0$ might be the control distribution cells and $p_1$ the distribution of cells after some chemical treatment. Flow matching constructs an interpolant object $x_t$ conditioned on data samples $x_0 \sim p_0$ and $x_1 \sim p_1$,

$$x_t = \alpha_t x_0 + \beta_t x_1, \quad t \in [0, 1], \tag{2}$$

where $\alpha_t, \beta_t$ are differentiable functions which define the interpolant path, satisfying $\alpha_0 = \beta_1 = 1, \alpha_1 = \beta_0 = 0$. A core insight of flow matching is that to learn the *unconditional* velocity field $v_t^\theta(x)$, parametrized by a neural network, it suffices to regress against the *conditional* target velocity $v_t(x_t|x_0, x_1) := \partial_t x_t = (\partial_t \alpha_t)x_0 + (\partial_t \beta_t)x_1$ over all data pairs observed during training:

$$\mathcal{L}_v(\theta) = \mathbb{E}_{(x_0,x_1)\sim p_{\mathrm{data}}, t\sim U(0,1)}||v_t^\theta(x_t) - v_t(x_t|x_0, x_1)||^2. \tag{3}$$

## 2.2 DATA SPARSITY IN DISTRIBUTION-TO-DISTRIBUTION LEARNING

Equation 2 exposes a key challenge in two-sided distribution learning with finite data: supervision via $x_t$ arrives only along sparse, one-dimensional interpolant paths determined by a limited set of data samples $x_0$ and $x_1$. Intuitively, in high dimensions, these thin supervision 'tubes' cover a vanishing fraction of $\mathcal{X}$. As $d$ grows or as the dataset shrinks, the learned $v_t^\theta$ must extrapolate more aggressively, degrading sample quality.

We make this challenge concrete with a synthetic task, CONCENTRICSHELLS, where the source and target distributions are $d$-dimensional concentric hyperspheres. The cost-minimizing transport moves points *radially outward* from the inner to outer shell. We train flow models from *unpaired* samples and evaluate two metrics: cosine similarity and sinkhorn distance. The **cosine similarity** between a source sample $x_0$ and the generated output $\hat{x}_1$ should be close to 1 for the CONCENTRIC-SHELLS geometry. The **sinkhorn distance**, an entropic-regularized Wasserstein proxy between generated and target samples, should be minimized.

Systematic experiments, carefully controlling the availability of training supervision, show that sparsity significantly degrades the quality of the learned transformation. **Sparsity hurts as data dimension grows**: Figure 3a shows that as $d$ scales from 2 to 2048, the cosine similarity falls from 0.98 to 0.77 while the sinkhorn distance rises from 0.3 to 4.7. **Sparsity hurts when data are few**: complementarily, Figure 3b varies the number of training examples from 128 to 8192 while fixing $d = 512$, showing that both metrics degrade sharply as supervision thins.

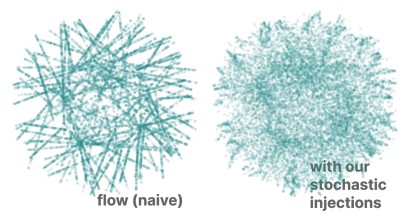

Figure 2: Our stochastic injections enable denser supervision, as visualized for $d = 2$ CONCENTRICSHELLS.

**Stochastic injections mitigate sparsity.** Our proposed stochastic injections substantially stabilizes learning under both stressors, making flow matching more robust to the sparsity induced by the curse of dimensionality. Figure 3 (blue curves) shows that with this intervention, the cosine alignment remains high across dimensions and number of data, and the sinkhorn distance is significantly reduced, with the largest gains precisely in the sparsity-challenged regimes of large $d$ and few samples where standard flow matching is most challenged. In the following section, we introduce our stochastic injections and explain why how they densify the training signal, leading to better distribution-to-distribution learning.

---

[1] In the one-sided setting with Gaussian $p_0$, flow matching is equivalent to $v$-prediction diffusion (Salimans & Ho, 2022).

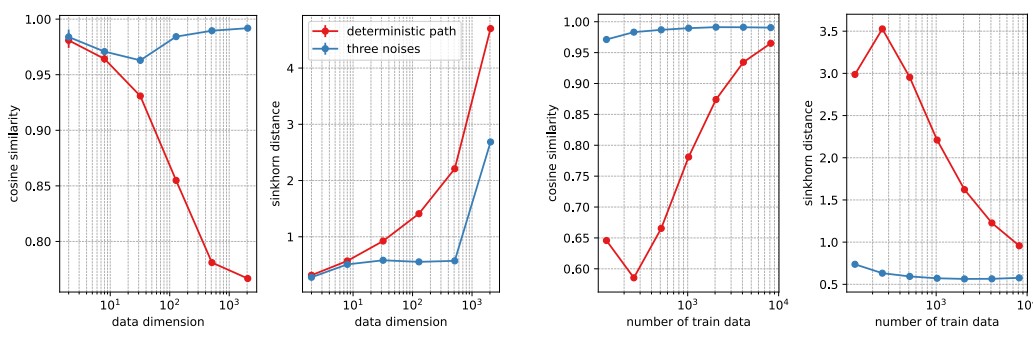

(a) Scaling data dimension.

(b) Scaling number of training data.

Figure 3: Sparsity stress tests on CONCENTRICSPHERES demonstrate that flow matching struggles in high dimensions and with few training examples. Our stochastic injections help 'de-sparsify' the supervision for the flow model, learning a more robust, generalizable velocity field.

## 3 METHOD

To counteract the challenges of sparse supervision highlighted in Section 2.2, we introduce three *stochastic injections* that densify training coverage with minimal code changes and almost no computational overhead.

### 3.1 TWO-STAGE TRANSFER LEARNING

**Problem.** In one-sided distributional learning, we can sample infinitely many $x_0 \sim p_0 = \mathcal{N}(0, I_d)$ from the source distribution for training, so the supervision is dense. In contrast, the two-sided setting relies on finite data samples from *both source and target* distributions, compounding the sparsity challenge.

**Solution.** Inspired by this insight and the success of transfer learning, we bootstrap from the supervision-abundant noise-to-target setting to our sparsity-challenged source-to-target setting. For the first stage of training (defined by a fixed number of epochs), we draw samples from $(x_0, x_1) \sim p(x_0, x_1)$ but discard the source $x_0$, supervising the velocity field with the interpolant

$$x_t = \alpha_t z + \beta_t x_1, \quad z \in \mathcal{N}(0, I_d). \tag{4}$$

In the second stage, we continue to draw samples $(x_0, x_1) \sim p(x_0, x_1)$, but fine-tune the same velocity field with now the interpolant

$$x_t = \alpha_t x_0 + \beta_t x_1. \tag{5}$$

The first stage learns vector fields that flow on $p_1$, without suffering the two-sided sparsity. The second stage can therefore adapt those fields to condition on $p_0$ described by data samples. In summary, pre-training on the noise-to-target setting supplies dense supervision for a flow that samples $p_1$, making the subsequent source-to-target fine-tuning stage more robust and sample-efficient.

### 3.2 PERTURBING THE SOURCE DISTRIBUTION

**Problem.** In the two-sided setting, both source and target distributions are observed only through finite data. Concretely, the empirical source distribution is a sum of Dirac masses centered on training examples. This discreteness poses two challenges: 1) the model may fail to generalize to unseen source samples, and 2) even if the ground-truth flow is learned, the recovered target remains a discrete mixture rather than a continuous distribution. We formalize this in the following lemma.

**Lemma 1.** *If the source distribution $p_0(x_0)$ is a mixture of delta distributions $\frac{1}{n}\sum_{i=0}^{n}\delta(x - x_i)$ with sample size $n$, then the ground-truth probability-flow ODE can only recover a mixture of delta distributions with sample size $n$.*

Proof is in Appendix B. The lemma implies that if source samples are sparser than target samples, the denser target distribution cannot be recovered. Moreover, training solely on sparse source data introduces generalization error when new source samples appear at test time.

**Solution.** To address the finite-support limitation of $p_0$, we densify the source distribution by injecting Gaussian perturbations. During training, we draw source and target samples from data, $(x_0, x_1) \sim p(x_0, x_1)$, and jitter the source sample to

$$\tilde{x}_0 = x_0 + z, \quad z \in \mathcal{N}(0, I_d) \tag{6}$$

with $z$ independently sampled every batch. The flow path now becomes

$$x_t = \alpha_t \tilde{x}_0 + \beta_t x_1. \tag{7}$$

Notably, we do *not* perturb $x_1$, as this causes the model to learn to learn a "noisy" target manifold. In summary, injecting Gaussian noise into each source sample densifies the support of the empirically observed $p_0$, enabling better generalization and recovery of the target distribution.

### 3.3 PERTURBING THE INTERPOLANT

**Problem.** With the deterministic interpolant of Equation 2, supervision lies on one-dimensional lines sampled during training. The direct interpolation between sparse sets of points – namely, the training examples from source and target – can only result in sparse interpolating sets. They have vanishing coverage in high dimensions, and training on them can lead to poor generalization.

**Solution.** To densify the interpolating points between each source-target pair sampled during training, we leverage *stochastic* interpolants that preserve the same marginal distributions at $t = 0$ and $t = 1$. Introduced by Albergo et al. (2023a), this framework generalizes the flow interpolant object (Equation 2) to

$$x_t = \alpha_t x_0 + \beta_t x_1 + \gamma_t z, \quad x_0 \in p_0, x_1 \in p_1, z \in \mathcal{N}(0, I_d), t \in [0, 1] \tag{8}$$

where $\gamma_t$ is a differentiable functions satisfying $\gamma_0 = \gamma_1 = 0$. A similar objective as flow matching (Equation 3) is employed to learn the velocity field $v_t^\theta$,

$$\mathcal{L}_v(\theta) = \mathbb{E}_{(x_0,x_1) \sim p_{\text{data}}, z \sim \mathcal{N}(0,I), t \sim U(0,1)} ||v_t^\theta(x_t) - v_t(x_t|x_0, x_1, z)||^2, \tag{9}$$

and to perform inference by numerically solving Equation 2. Additionally, stochastic interpolants support SDE sampling with a score field that models $s_t(x) = \nabla \log p_t(x)$.[2] It may be shown that, for every $t$ where $\gamma_t \neq 0$,

$$s_t(x) = -\gamma_t^{-1} \mathbb{E}(z|x_t = x). \tag{10}$$

Similarly to learning the velocity field, a neural-network-parametrized $s_t^\phi$ may be learnt by regressing against $z$ over the interpolants observed during training. For numerical stability, we often choose the parametrization $\eta_t(x) = \gamma_t s_t(x)$ and learn

$$\mathcal{L}_s(\phi) = \mathbb{E}_{(x_0,x_1) \sim p_{\text{data}}, z \sim \mathcal{N}(0,I), t \sim U(0,1)} ||\eta_t^\phi(x_t) - \eta_t(x_t|x_0, x_1, z)||^2. \tag{11}$$

At inference, we can choose an arbitrary diffusion schedule $\sigma_t$ and numerically integrate with an SDE solver, such as Euler-Maruyama:

$$dx_t = \left(v_t(x_t) - \frac{1}{2}\sigma_t^2 \gamma_t^{-1} \eta_t(x_t)\right) dt + \sigma_t dW_t, \tag{12}$$

where $W_t$ is the Wiener process.

Injecting stochasticity into the interpolant *densifies* the distribution of the interpolating points, tightening the discrepancy between empirical samples and population. We formalize this intuition with the following theorem.

**Theorem 1** (Informal). *Let $\mathcal{L}_{FM}(\theta, t), \mathcal{L}_{SI}(\theta, t)$ be the population risk of flow matching and our stochastic injection loss at time $t$, and $\hat{\mathcal{L}}_{FM}(\theta, t), \hat{\mathcal{L}}_{SI}(\theta, t)$ be their empirical risks with $n$ i.i.d. samples. Let $p_t(x_t), q_t(x_t)$ be the respective population distribution of $x_t$, and $\hat{p}_t(x_t), \hat{q}_t(x_t)$ be their empirical distributions, the 1-Wasserstein distance $\mathbb{W}_1(p_t, \hat{p}_t), \mathbb{W}_1(q_t, \hat{q}_t)$ characterizes each loss' generalization gap. Moreover, $\mathbb{W}_1(q_t, \hat{q}_t) \leq \mathbb{W}_1(p_t, \hat{p}_t)$.*

---

[2]If $p_0$ or $p_1$ is Gaussian, i.e. the one-sided setting, the score can be directly obtained from $v_t$ by the relation $v_t(x) = \frac{\dot{\beta}_t}{\beta_t}x - \gamma_t(\dot{\gamma}_t - \frac{\dot{\beta}_t \gamma_t}{\beta_t})s_t(x)$, without separately learning the score.

See Appendix B for a more formal statement and proof. This theorem allows us to use the 1-Wasserstein distance to approximately measure the generalization gap, and shows that the gap is smaller if $x_t$ is a stochastic interpolation. In summary, perturbing the interpolant path densifies supervision along the interpolating path, tightening the generalization gap.

**Designing the interpolant noise schedule.** Stochastic interpolants extend the design space to the choice of $\gamma_t$. Prior work (Albergo & Vanden-Eijnden, 2022) favored $\gamma_t = \sqrt{2t(1-t)}$, which maintains identical variance $\alpha_t^2 + \beta_t^2 + \gamma_t^2 = 1$ over all timesteps, assuming linear path and $p_0, p_1$ normalized to unit variance. However, this does not guarantee optimality in real data distributions; in fact, the divergence of $\partial_t \gamma_t$ at the endpoints creates numerical instability when regressing the conditional target velocity. In this work, we explore the shape and scale of $\gamma$ on a real world image two-sided image distribution, considering three noise schedules,

$$\gamma_t = \begin{cases} a\sqrt{2t(1-t)} & \text{square-root} \\ a\sin^2(\pi t) & \text{sin-squared} \\ a\, t(1-t) & \text{quadratic} \end{cases} , \quad a \in \mathbb{R}^+, \tag{13}$$

where $a$ controls the noise scale. Since our work focuses on the impact of stochastic injection, we fix $\alpha_t = 1 - t$ and $\beta_t = t$, the conditional optimal transport between two Gaussians.

### 3.4 MODEL

Algorithm 1 summarizes the flow matching training with all three stochastic injections. We implement both the velocity and score fields with a tied UNet (Rombach et al., 2021), using a shared backbone and a lightweight projection head that maps the feature space to $v_\theta$ and $\eta_\phi$ separately. We optimize the combined objective $\mathcal{L}_v(\theta) + \mathcal{L}_\eta(\phi)$. Following common practice in image generation, we perform training and inference in the latent space of a variational autoencoder (VAE) which reduces data dimension and compute.

## 4 EXPERIMENTS

### 4.1 DATASETS

We demonstrate the efficacy of our method on five tasks representing a wide spectrum of scientific and natural image domains, modeling cellular response, seasonal transitions, disease progression, and galaxy evolution. On **BBBC**, a cell microscopy dataset, we learn chemically-conditioned morphological changes, where cells under the control condition form the source distribution and cells post-intervention form the target distribution. On **SEASONET**, which comprises satellite images covering Germany over four seasons, and on **YOSEMITE**, which comprises user-uploaded images of Yosemite National Park, we learn to map images from the summer to the same view in winter. On **MIMIC-CXR**, we learn the markers of Pleural Effusion (PE) with chest radiographs of PE-negative patients forming the source distribution and those of PE-positive patients forming the target distribution. On **GALAXIESML**, we learn cosmological evolution reflected in galaxy images, transforming low redshift images onto high redshift ones. See Appendix C for further details.

### 4.2 RESULTS

**Baselines.** We evaluate our proposed stochastic injections by comparing to standard **flow matching**, as well as several other unpaired distribution-to-distribution learning methods. **UNSB** (Kim et al., 2023) leverages a multi-step GAN to learn a Schrödinger bridge variant between the source and target. **DDIB** (Su et al., 2022) learns to transform each source sample to an intermediate Gaussian latent and then to the generated target sample. **SDEdit** (Meng et al., 2021) partially noises the source sample and denoises the remaining timesteps into a target sample.

**Improved distribution learning.** Table 1 shows that across all datasets, the stochastic injections lead to significant improvements over standard flow matching, averaging 13 FID points, while also outperforming UNSB, DDIB, and SDEdit baselines by 9 FID points. Qualitative examples in Figure 4 show that our generated samples are highly realistic while preserving the structure of the source

Table 1: Frechet Inception Distance (FID) (Heusel et al., 2017) of target samples on the held out test set demonstrate that our method significantly improves standard flow matching and outperforms baselines across five diverse image datasets.

|  | BBBC | SeaoNet | Yosemite | MIMIC-CXR | GalaxiesML |
|---|---|---|---|---|---|
| UNSB | 94.6 | 89.6 | 73.9 | 45.0 | 178.1 |
| DDIB | 30.0 | 71.4 | 84.7 | 30.9 | 10.0 |
| SDEdit | 164.1 | 299.7 | 261.7 | 293.0 | 136.9 |
| Flow (standard) | 33.6 | 80.0 | 87.3 | 34.9 | 13.1 |
| Flow w. stochastic (ours) | **19.9** | **60.5** | **71.5** | **22.9** | **7.4** |

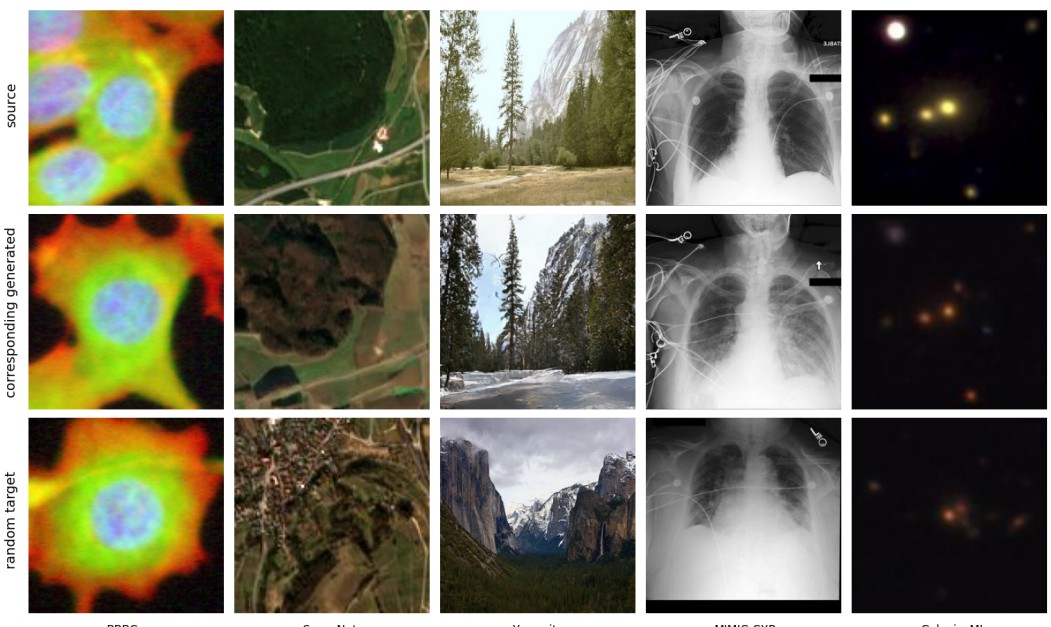

Figure 4: Qualitative examples. The top row is from the source side of the test set. The middle row is the corresponding model generation with the top image as the source, using a flow model trained with all three forms of stochastic injection. The bottom row is a random sample from the target side of the test set.

image. For example, the BBBC column shows a conditional generation with the floxuoridine intervention, which inhibits DNA replication. The generated sample exhibits the expected reduced density while maintaining the position of the central cell's nucleus and surrounding cytoskeleton. The MIMIC-CXR example shows blunting of the costophrenic angle characteristic of pleural effusion, while preserving the patient's orientation.

**Improved source-target correspondence.** Preserving the content of the source image and modifying only the characteristics specific to the true transformation is desirable because it highlights the true transformation of interest. In our setting, this correspondence between source and target correlates to the transport cost. Though flow models do not solve an optimal transport problem, we observe that implicit regularization results in a visual correlation between each source sample and its generated target. Furthermore, our stochastic injections *reduce the transport costs* of standard flow matching. This is reflected in two metrics: the pixel-space mean-squared-error (MSE) between input source image and generated target.[3], and the percentage of source samples that are *matched* to

---

[3]Pixel values are normalized to the range $[-1, 1]$ and the reported MSE number is averaged over the image dimension and across all pairs.

Table 2: The stochastic injections generally improve the alignment between the source sample and the generated target sample, compared to standard (fully deterministic) flow matching.

|  |  | BBBC | SeasoNet | Yosemite | MIMIC-CXR | GalaxiesML |
|---|---|---|---|---|---|---|
| MSE ($\downarrow$) | determ. | **0.055** | 0.042 | 0.060 | 0.025 | 0.0054 |
|  | stoch. | 0.057 | **0.032** | **0.041** | **0.014** | **0.0046** |
| % matched ($\uparrow$) | determ. | **24%** | 39% | 63% | 54% | 13% |
|  | stoch. | **24%** | **54%** | **71%** | **74%** | **47%** |

Table 3: Ablations on two datasets support that each form of stochastic injection is independently beneficial for distribution learning, and the best model uses a all three. All numbers are test FID.

|  | all noises | no two-stage | no src noise | no interp. noise | no noises |
|---|---|---|---|---|---|
| BBBC | **19.9** | 20.6 | 27.9 | 23.2 | 33.6 |
| SEASONET | **60.5** | 62.8 | 62.2 | 76.0 | 80.0 |

their corresponding generated sample by linear sum assignment.[4] Table 2 shows that the stochastic injections reduces MSEs by 22% and improves assignment matches by 15 percentage points, indicating stronger alignment between source target to better highlight the true effect of the underlying transformation.

## 4.3 ABLATIONS

**Each form of stochastic injection helps.** Table 3 shows that removing any of the three stochastic injections degrades performance: all are necessary to achieve the strongest FIDs. Their relative contributions depend on the data distribution. For example, perturbing the source distribution is especially valuable on BBBC, while the stochastic interpolant drives most of improvement on SEASONET.

**Interpolant noise schedule.** We experiment with several choices of $\gamma_t$ on BBBC. Figure 5a shows that the sine-squared noise schedule achieves the strongest improvement over the deterministic baseline. Interestingly, performance is degraded by the square-root noise schedule favored in prior work, possibly due to numerical instability as discussed in Section 3.3. The scale of the noise schedule ($a$ in Equation 13) is also important: Figure 5b suggests that too low of a noise scale underutilizes the benefit of this stochastic injection. Guided by these results, we fixed $\gamma_t$ to be the sine-squared schedule at scale 1.0 for the main experiments. We found that these are reasonable choices to achieve strong gains over the deterministic interpolant, but acknowledge that optimal choices require dataset-specific tuning.

**Sampling strategy.** Stochastic interpolants ($\gamma_t \neq 0$) support both ODE and SDE sampling – which is better? Consistent with Ma et al. (2024)'s observations, Figure 5c shows that SDE surpasses ODE in the limit of many inference steps. However, we find that such gains can be obtained with a simpler technique: adding Gaussian noise to each source sample. With this inference-time noising, we find that SDE sampling no longer offers any gains over ODE. Since ODEs are cheaper to simulate numerically, require fewer inference steps, and do not require learning score field, we reported our main results with ODE-generated samples with inference-time source noising as described.[5]

---

[4]We use pixel-space Euclidean distance as the cost metric for the linear assignment, with the set of source images forming one side of the bipartite graph and the set of generated target images forming the other side. Intuitively, a score of 100% means that each source image is most similar in pixel space to its corresponding generated sample.

[5]The deterministic flow baseline is sampled without this inference-time Gaussian noise, as it is OOD for the model and significantly degrades sample quality.

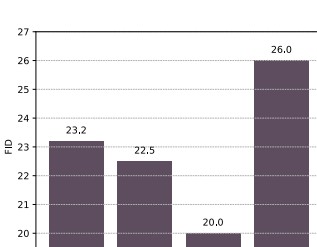 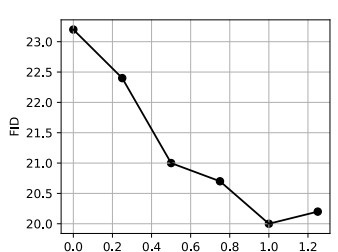 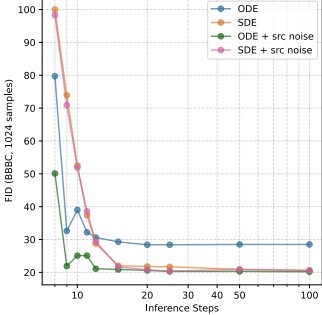

(a) The sine-squared interpolant noise schedule outperforms quadratic and sqrt schedules.

(b) The noise scale $a$ (shown for the sine-squared noise schedule) can be tuned for best performance.

(c) At inference, ODE sampling with Gaussian perturbations to the source gives best results.

Figure 5: Ablations suggest that the sine-squared function (Figure 5a) at scale $a = 1.0$ (Figure 5b) is the best choice of interpolant noise schedule $\gamma$. For inference, we find that ODE sampling with source noising performs better than SDE sampling. All metrics are FID are reported on the test set of BBBC.

## 5 RELATED WORK

**Distribution-to-distribution learning.** Earlier works (Zhu et al., 2017; Park et al., 2020) leveraged GANs with cycle consistency loss to learn transformations between two image distributions. More recently, (Kim et al., 2023) designs a multi-step GAN with optimal transport regularization. However, GANs can suffer from mode collapse; we focus on improving probability flow models, which have become the dominant paradigm for scalable and stable generative modeling. Liu et al. (2023) and Delbracio & Milanfar (2023) learn diffusion bridges to transform between two distributions, but require access to paired data. Tong et al. (2023) and De Bortoli et al. (2021) aim to learn optimal transport (OT) maps, but rely on noisy approximations to the true OT pairings and scale poorly to high-dimensional data. Other works adapt traditional noise-to-target diffusion to learn distribution-to-distribution transformations: Su et al. (2022) by concatenating two diffusion models and Meng et al. (2021) by denoising from the source samples at an intermediate timestep. In contrast, we directly learn the transformation the source and target distribution, and show that this achieves superior sample quality.

**Stochastic interpolants.** Albergo & Vanden-Eijnden (2022) and Albergo et al. (2023a) introduce stochastic interpolants (SI), unifying flows and diffusion under a common framework for bridging two data distributions. Albergo & Vanden-Eijnden (2022) shows that the stochasticity of the interpolant can improve learning on toy distributions. Albergo et al. (2023b) uses the language of SI for image translation with paired data, but chooses a deterministic interpolant equivalent to the usual flow matching. Ma et al. (2024) explores the diffusion design space using the SI framework, but does not study general source-to-target transformations. Höllmer et al. (2025) leverages stochastic interpolants for the design of inorganic crystalline materials. Concurrent work Singh & Lagun (2025) learns interpolants in conjunction with a VAE. Despite recent interest, we are still in the nascent stages of understanding SI's practical utility in their full generality beyond flow and diffusion special cases. Our work demonstrates that perturbing the interpolant path, in conjunction with other stochastic injection techniques, improves distribution-to-distribution learning.

## 6 CONCLUSION

In this work, we explore flow models for unpaired distribution-to-distribution learning, and identify *sparse supervision from finite data* as a core challenge impeding performance. Our proposed stochastic injections alleviate this data sparsity, lowering FIDs by 13 points relative to standard flow matching and 9 points relative to other baselines, across five imaging datasets spanning biology, satellite data, health, and astronomy. Our contributions make flow matching a powerful method for modeling diverse distributional transformations in science.

ETHICS STATEMENT

Our research is primarily methodological in nature and does not raise ethical concerns regarding data privacy, fairness, or potential misuse.

REPRODUCIBILITY STATEMENT

We provide the code to reproduce our experiments at `https://anonymous.4open.science/r/StochasticInjections-702D`.

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

## A  ALGORITHM

---

**Algorithm 1** Training

---

**Require:** Training data $\{x_0^{\}}$ and $\{x_1\}$, epochs $E_{\text{pre-train}}$, $E_{\text{total}}$, interpolant functions $\alpha_t, \beta_t, \gamma_t$, interpolant noise scale $a$, untrained velocity field $v_\theta$ and score field $\eta_\phi$

1: **for** epoch $= 1$ to $E_{\text{pre-train}}$ **do**         ▷ Stage 1: pre-train on noise-to-target
2:  **for** each batch **do**
3:   Sample $x_0, x_1, t \sim \mathcal{U}(0, 1)$
4:   $z_{\text{two-stage}} \sim \mathcal{N}(0, I)$
5:   $z_{\text{interpolant}} \sim \mathcal{N}(0, I)$
6:   $x_t = \alpha_t z_{\text{two-stage}} + \beta_t x_1 + \gamma_t z_{\text{interpolant}}$      ▷ Perturb the interpolant
7:   $v_t(x_t|x_0, x_1, z) = \dot{\alpha}_t z_{\text{two-stage}} + \dot{\beta}_t x_1 + \dot{\gamma}_t z$
8:   $\eta_t^*(x_t|x_0, x_1, z) = z_{\text{interpolant}}$
9:   Update $\theta, \phi$ on $\mathcal{L}_v(\theta) + \mathcal{L}_\eta(\phi)$
10:  **end for**
11: **end for**
12: **for** epoch $E_{\text{pre-train}}$ to $E_{\text{total}}$ **do**        ▷ Stage 2: fine-tune on source-to-target
13:  **for** each batch **do**
14:   Sample $x_0, x_1, t \sim \mathcal{U}(0, 1)$
15:   $z_{\text{source}} \sim \mathcal{N}(0, I)$
16:   $z_{\text{interpolant}} \sim \mathcal{N}(0, I)$
17:   $\tilde{x}_0 = x_0 + z_{\text{source}}$          ▷ Perturb the source sample
18:   $x_t = \alpha_t \tilde{x}_0 + \beta_t x_1 + \gamma_t z_{\text{interpolant}}$      ▷ Perturb the interpolant
19:   $v_t(x_t|x_0, x_1, z) = \dot{\alpha}_t \tilde{x}_0 + \dot{\beta}_t x_1 + \dot{\gamma}_t z$
20:   $\eta_t^*(x_t|x_0, x_1, z) = z$
21:   Update $\theta, \phi$ on $\mathcal{L}_v(\theta) + \mathcal{L}_\eta(\phi)$
22:  **end for**
23: **end for**

---

**Algorithm 2** Inference

---

**Require:** Source sample $x_0$, learned fields $v_\theta, \eta_\phi$, interpolant noise schedule $\gamma_t$, inference source noise scale $\epsilon$, diffusion coefficient $\sigma_t$, step size $\Delta t$, numerical solver function $\texttt{Step}$

1: $z \sim \mathcal{N}(0, I)$
2: $\tilde{x}_0 = x_0 + \epsilon z$
3: Initialize $x_t = \tilde{x}_0$ at $t = 0$
4: **for** $t = 0$ to $1$ **do**
5:  $x_{t+\Delta t} = \texttt{Step}(x_t, t, v_\theta, \eta_\theta, \gamma_t, \sigma_t)$
6: **end for**

---

## B  THEOREMS AND DERIVATIONS

To formally state the theorem, we need to first define the population risk and empirical risk for flow matching and our loss. Generally, both losses can be written as

$$\mathcal{L}_{\text{FM}}(\theta, t) = \mathbb{E}_{x_t \sim p_t(x_t)} \left[ \frac{1}{2} \|v_t^\theta(x_t) - v_t^{\text{FM}}(x_t)\|^2 \right] \tag{14}$$

$$\mathcal{L}_{\text{SI}}(\theta, t) = \mathbb{E}_{x_t \sim q_t(x_t)} \left[ \frac{1}{2} \|v_t^\theta(x_t) - v_t^{\text{SI}}(x_t)\|^2 \right] \tag{15}$$

where $v_t^*(x_t)$ is the marginal velocity at $x_t$, and $p_t(x_t), q_t(x_t)$ are the ground-truth distributions to draw $x_t$ for each loss.

**Theorem 2.** *Let $\mathcal{L}_{FM}(\theta, t), \mathcal{L}_{SI}(\theta, t)$ be the population risk of Flow Matching and our stochastic injection loss at $t \in [0, 1]$, and $\hat{\mathcal{L}}_{FM}(\theta, t), \hat{\mathcal{L}}_{SI}(\theta, t)$ be their empirical risks with $n$ i.i.d. samples. Let*

$p_t(x_t), q_t(x_t)$ be the respective population distribution of $x_t$, and $\hat{p}_t(x_t), \hat{q}_t(x_t)$ be their empirical distributions, and assume each loss is Lipschitz continuous $x_t$, we have

$$|\mathcal{L}_{FM}(\theta, t) - \hat{\mathcal{L}}_{FM}(\theta, t)| \leq L\mathbb{W}_1(p_t, \hat{p}_t), \qquad |\mathcal{L}_{SI}(\theta, t) - \hat{\mathcal{L}}_{SI}(\theta, t)| \leq L\mathbb{W}_1(q_t, \hat{q}_t) \qquad (16)$$

for some constant $L$. Moreover, $\mathbb{W}_1(q_t, \hat{q}_t) \leq \mathbb{W}_1(p_t, \hat{p}_t)$.

*Proof.* Without loss of generality we let $\mathcal{G}_\theta^{FM}(x_t) = \frac{1}{2}\|v_t^\theta(x_t) - v_t^{FM}(x_t)\|^2$ with Lipschitz constant $L_{FM}$, and similarly $\mathcal{G}_\theta^{SI}(x_t)$ has Lipschitz constant $L_{SI}$, and so

$$\mathcal{L}_{FM}(\theta, t) = \mathbb{E}_{x_t \sim p_t(x_t)}[\mathcal{G}_\theta(x_t)], \quad \mathcal{L}_{SI}(\theta, t) = \mathbb{E}_{x_t \sim q_t(x_t)}[\mathcal{G}_\theta(x_t)] \qquad (17)$$

By Kantorovich–Rubinstein duality of $\mathbb{W}_1(p_t, \hat{p}_t)$, we have

$$\left|\mathcal{L}_{FM}(\theta, t) - \hat{\mathcal{L}}_{FM}(\theta, t)\right| = \left|\mathbb{E}_{x_t \sim p_t(x_t)}[\mathcal{G}_\theta(x_t)] - \mathbb{E}_{x_t \sim \hat{p}_t(x_t)}[\mathcal{G}_\theta(x_t)]\right| \qquad (18)$$

$$\leq L_{FM} \sup_{f \in \mathcal{F}} \left|\mathbb{E}_{x_t \sim p_t(x_t)}[f(x_t)] - \mathbb{E}_{x_t \sim \hat{p}_t(x_t)}[f(x_t)]\right| \qquad (19)$$

$$\leq \max\{L_{FM}, L_{SI}\} \sup_{f \in \mathcal{F}} \left|\mathbb{E}_{x_t \sim p_t(x_t)}[f(x_t)] - \mathbb{E}_{x_t \sim \hat{p}_t(x_t)}[f(x_t)]\right| \quad (20)$$

$$= L\mathbb{W}_1(p_t, \hat{p}_t) \qquad (21)$$

where we let $L = \max\{L_{FM}, L_{SI}\}$ and $\mathcal{F}$ is a function set with Lipschitz constant of at most 1. The same conclusion holds for $\mathcal{L}_{SI}(\theta, t)$.

For the final result, since $q_t(x_t)$ is determinstic interpolation (drawn from $p_t(x_t)$) with Gaussian noise, it can be equivalently written as $q_t = p_t * N_t$ where $N_t(x) = \mathcal{N}(0, \sigma_t^2 I)$, a Gaussian convolution of $p_t(x_t)$ with variance $\sigma_t^2$. Similarly $\hat{q}_t = \hat{p}_t * N_t$. Therefore,

$$\mathbb{W}_1(q_t, \hat{q}_t) = \mathbb{W}_1(p_t * N_t, \hat{p}_t * N_t) \overset{(i)}{\leq} \mathbb{W}_1(p_t, \hat{p}_t) \qquad (22)$$

where (i) is due to Wasserstein-reducing property of Gaussian smoothing (Chen & Niles-Weed, 2022). □

We remark that $\mathbb{W}_1(p_t, \hat{p}_t)$ is a measure of an upper bound of the generalization gap, and it does not strictly characterize the gap, so the exact relationship between the two generalization gaps cannot be measured precisely. However, we use the 1-Wasserstein distance as an approximation of the gap to give a rough intuition on why injecting stochastic noise can help test performance.

**Lemma 1.** *If the source distribution $p_0(x_0)$ is a mixture of delta distributions $\frac{1}{n}\sum_{i=0}^{n}\delta(x - x_i)$ with sample size $n$, then the ground-truth probability-flow ODE can only recover a mixture of delta distributions with sample size $n$.*

*Proof.* We know that the ground-truth probability-flow ODE path does not cross, and therefore the ground-truth ODE flow is a one-to-one function. Let $\Phi(x_0)$ denote the ground-truth flow path following probability-flow ODE, and consider the pushforward distribution via the flow path as $(\Phi \# p_0)(x_1) = \frac{1}{n}\sum_{i=0}^{n}\Phi\#\delta(x_0 - x_i) = \frac{1}{n}\sum_{i=0}^{n}\delta(x_1 - \Phi(x_i))$, which is a mixture of delta distributions with sample size $n$. □

## C DATASETS

We describe each dataset below.

- BBBC. We use the BBBC021v1 image set (Caie et al., 2010) from the Broad Bioimage Benchmark Collection (Ljosa et al., 2012). This dataset contains fluorescent microscopy of cells treated with 26 small molecule chemicals, forming a conditional target distribution for each chemical. Three color channels correspond to DNA, F-actin, and beta-tubulin markers.

- SEASONET. This dataset contains multi-spectral aerial image patches covering the surface of Germany from the Sentinel-2 mission, collected from April 2018 to February 2019 (Koßmann et al., 2022). The images are available in standard RGB channels and sorted into each of four seasons. We use only the summer and winter splits for the source and target distributions respectively.

Table 4: Dataset statistics.

|  | # train(A) | # train(B) | # test(A) | # test(B) | resolution | domain |
|---|---|---|---|---|---|---|
| BBBC | 63,781 | 6,210 | 690 | 7,119 | $256 \times 256$ | cell microscopy |
| SeasoNet | 235,826 | 104,432 | 1,024 | 1,024 | $120 \times 120$ | satellite |
| Yosemite | 1,231 | 962 | 309 | 238 | $256 \times 256$ | natural |
| MIMIC-CXR | 16,038 | 44,372 | 1,024 | 1,024 | $256 \times 256$ | medical x-ray |
| GalaxiesML | 35,725 | 45,741 | 1,024 | 1,024 | $127 \times 127$ | astronomy |

- YOSEMITE. We use images of Yosemite National Park collected by Zhu et al. (2017) via the Flickr API. The dataset separates images taken in the summer and images taken in the winter, which we use as the source and target distributions.

- MIMIC-CXR. MIMIC-CXR is a medical imaging dataset of chest radiographs (Johnson et al., 2019). We filter to those images with the antero-posterior view angle. Pleural effusion is a condition characterized by fluid around the lungs. The source distribution is defined as scans from patients with pleural effusion value of 0.0, and the target distribution scans from patients with pleural effusion value of 1.0. The scans are in single-channel grayscale. We resize them to $256 \times 256$.

- GALAXIESML. We use galaxy images from the Hyper-Suprime-Cam (HSC) Survey (Aihara et al., 2019) as processed by Do et al. (2024). This dataset contains five photometric bands (g, r, i, z, y) as well as spectroscopically confirmed redshifts. We use the g, r, and i channels to construct a 3-channel image. The images with redshift values 0.3-0.5 are used as the source distribution and images obtained at redshift values 0.5-0.7 are used as the target distribution.

Datasets statistics are summarized in Table 4. All FID numbers are reported over the held out test sets.

## D    TRAINING DETAILS

### D.1    EXPERIMENTS ON CONCENTRICSHELLS

In this task, the source distribution (inner shell) is a hypersphere centered at the origin with radius 1 and the target distribution (outer shell) is a hypersphere centered at the origin with radius 2. For both the source and target distributions, each sample is obtained by sampling over the $d$-dimensional shell, then perturbing this with a random normal noise component with standard deviation 0.1.

For the data dimension scaling experiment, each training run operates over 1024 samples from the source distribution (inner shell) and 1024 samples from the target distribution (outer shell). For the dataset size scaling experiment, the data dimension is fixed to 512. We use the Adam optimizer at learning rate 0.01 and batch size 256. The velocity field is fit by a simple 4-layer MLP with hidden dimension 64 and an ELU non-linearity between each fully connected layer.

All metrics are reported over a test set of 512 samples. We compute the Sinkhorn distance with entropy regularization of 0.1.

### D.2    VARIATIONAL AUTOENCODER

We learn velocity and score fields in the latent space of a VAE for each of the image datasets introduced in Section 4.1. During training and inference, we use different variational autoencoders that is best adapted to each dataset. They are first trained from the same data as what is available for the distribution learning, and the VAE weights are subsequently frozen. We same architecture as the $f = 4$ autoencoder from Rombach et al. (2021). The VAE for BBBC is trained from scratch. The VAE for each of SEASONET, MIMIC-CXR, and GALAXIESML are fine-tuned from Rombach et al. (2021)'s kl-f4 checkpoint, trained for 176991 steps, at the default KL regularization penalty of $1e-6$. For YOSEMITE, we directly use their pre-trained autoencoder as we found fine-tuning on YOSEMITE led to overfitting and performance degradation, likely due to the small dataset size.

## D.3 MAIN EXPERIMENTS

On all datasets, models were trained with constant learning rate $1e-4$ with the AdamW optimizer (betas 0.9 and 0.95). We maintain an EMA-weighted copy of the model with 0.999 decay. For the conditional dataset BBBC, we drop class labels with probability 0.2. Each epoch iterates over all training data in the target distribution while randomly sampling training data in the source distribution. We train for 200, 100, 2000, 60, 80 epochs for each of the datasets BBBC, SEA-SONET, YOSEMITE, MIMIC-CXR, and GALAXIESML respectively, based on observed convergence. When training with the two-stage scheme, some fraction of these epochs are reserved for the noise-to-target stage. Hence the two-stage training does not incur additional compute.

Sampling is performed with the Heun solver with 50 inference steps (corresponding to 100 NFEs) unless stated otherwise. The stochastic variant of the Heun solver (Karras et al., 2022) was used for the ODE/SDE comparison experiments (Figure 5c). Similar to some prior work (Ma et al., 2024), we chose the diffusion coefficient $\sigma_t^2/2 = \sin^2(\pi t)$. We also experimented with a time-independent $\sigma_t$ but found it performed worse than a schedule that is tapered at the $t = 0$ and $t = 1$ endpoints. Additionally, we set the diffusion coefficient to 0 within a margin $\epsilon = 1e-3$ near the endpoints, to avoid the numerical instability caused by the factor of $\gamma^{-1}$. We find this is crucial to obtain reasonable samples.

For the DDIB and SDEdit baselines, we require access to a generative model that can conditionally flow from noise to both the source and the target. For full comparability, we train a noise-to-source/target flow matching model for each dataset, keeping all hyperparameters consistent with the flow baseline where applicable.

