# OpenReview forum: "Three Forms of Stochastic Injection for Improved Distribution-to-Distribution Generative Modeling"
_ICLR.cc/2026/Conference — ICLR 2026 Conference Withdrawn Submission_

### Official Review · Reviewer_osuf · 2025-11-01

**Soundness:** 3
**Presentation:** 3
**Contribution:** 2
**Rating:** 4
**Confidence:** 3

**Summary:**

The paper proposes a simple and computationally efficient three-stage method that injects stochasticity into the training process by perturbing source samples and flow interpolants.
Extensive experiment is performed to verify the effectiveness of the proposed.

**Strengths:**

1.	The proposed method is practical in real world with sparse supervision.

2.	The method is intuitionally by the demonstration of toy experiments.

**Weaknesses:**

1. The two-stage learning is a kind of transfer learning, while perturbing the source distribution is a form of data augmentation, and path interpolation perturbation has been explored in prior work. The novelty the paper is unclearly and please clarify the main contribution.

2. The current evaluation methodology lacks persuasiveness. To strengthen the validation, the model could be trained on unpaired datasets and then tested on paired datasets (e.g., the SynthRAD medical image paired dataset), reporting more convincing quantitative metrics such as SSIM and PSNR.


3. To demonstrate the generality of random injection, experiments are needed to validate its effectiveness in flow matching based method, such as RF and OT-CFM.



4. Please provide comparisons with more advanced unpaired baselines, such as OT-CFM for Unpaired Data Translation, Bi-DPM for partially paired data.


5. Why does perturbing the source distribution not have a hyperparameter for perturbation magnitude. Dose the impact of the perturb is negligible? The authors acknowledge that the perturbation magnitude is a parameter requiring fine-tuning.
Ablation study for the two perturbation magnitudes is necessary

[1] Alexander Tong and Kilian FATRAS and Nikolay Malkin and Guillaume Huguet and Yanlei Zhang and Jarrid Rector-Brooks and Guy Wolf and Yoshua Bengio. Improving and generalizing flow-based generative models with minibatch optimal transport. Transactions on Machine Learning Research, 2835-8856, 2024
[2] he Xiong and Qiaoqiao Ding and Xiaoqun Zhang, Bi-modality medical images synthesis by a bi-directional discrete process matching method. https://openreview.net/forum?id=GqsepTIXWy, 2025.

**Questions:**

In the left of Figure3(a), why does the blue curve first decrease and then increase?

---

### Official Review · Reviewer_SZP1 · 2025-11-01

**Soundness:** 3
**Presentation:** 4
**Contribution:** 3
**Rating:** 4
**Confidence:** 3

**Summary:**

This paper introduces three distinct noise-injection mechanisms designed to improve distribution learning. The proposed methods are accompanied by theoretical analysis and validated through numerical experiments.

**Strengths:**

The proposed noise-injection techniques are both simple and broadly applicable. Moreover, the paper provides a thorough and rigorous theoretical analysis that explains why these methods are effective. This combination of practical simplicity and strong theoretical grounding gives the work both immediate applicability and a solid conceptual foundation. Furthermore, the inclusion of numerical experiments to support the theoretical claims strengthens the paper, as it demonstrates that the methods perform well in practice.

**Weaknesses:**

One potential weakness or concern is that many of the noise-injection techniques proposed in the paper are not entirely new, even though the authors present them as novel contributions. I recommend that the authors provide a clear and fair survey of existing methods in the literature and emphasize the theoretical contributions of their work, which I believe offer the most significant novelty in this work.

**Questions:**

Could the authors include toy examples to illustrate the benefits of these techniques, particularly the second one, where Lemma 1 appears especially interesting?

---

### Official Review · Reviewer_htau · 2025-11-01

**Soundness:** 2
**Presentation:** 2
**Contribution:** 2
**Rating:** 2
**Confidence:** 3

**Summary:**

This paper proposed  three forms of stochastic injection to improve flow matching for distribution-to-distribution generative modeling. The method introduces randomness in transfer learning, source perturbation, and interpolant perturbation to address data sparsity and improve generalization. Experiments on five datasets show consistent FID improvements compared to standard flow matching and prior baselines.

**Strengths:**

1.The proposed stochastic injections (transfer pretraining, source perturbation, and interpolant noise) are conceptually intuitive and easy to integrate into existing flow matching pipelines. Numerical result show that empirical improvements across multiple datasets  in terms of FID.
2.Experiments span five datasets showing consistent improvements in FID and qualitative fidelity.

**Weaknesses:**

1.Although the method is stated as stochastic injection into flow matching method with three types of randomness, the  proposed method is a combination of three known techniques: transfer learning (training from gaussian-target), data augmentation (perturbation) and combining  with diffusion method training ( perturbing the interpolant).  Thus it is not surprising that it improves upon the basic flow matching method.  Also the motivation of the paper is to deal with generative ability from few examples, however this aspect is not explored how the proposed model deal with the scarce of data, not even empirical study how the model improves for different level of data.

2.The experimental comparisons in the paper is not very convincing. Not only some recent flow matching models such as CFM, Rectified are not included, and the image quality criteria with FID is not sufficient  for real image translation applications  The effectiveness of image translation method should be evaluated  either with image quality assessment (PSNR, SSIM) or with downstream task performance.   The paper also lacks runtime analysis, leaving uncertainty about the method’s efficiency in practice.

3. Overall the paper is poorly written with many flaws. For example,  in the beginning it is mentioned that the method is used with few samples with concentric examples, however the quantity of samples are not well present. Also what does it mean by standard flow matching method, as there is no specific reference?

**Questions:**

IN addition to the questions mentioned in the weakness, there are two more:
1.While the paper briefly examines the noise schedule and scale parameter aaa on the BBBC dataset, it remains unclear how sensitive the overall performance is to these hyperparameters. Could the authors provide a broader sensitivity analysis across datasets?
2.Can the stochastic injection strategy generalize to non-image modalities?

---

### Note · Authors · 2025-11-14

I have read and agree with the venue's withdrawal policy on behalf of myself and my co-authors.